# Shadow loss: Memory-linear deep metric learning with anchor projection

## Abstract

Deep metric learning objectives (e.g., triplet loss) require storing and comparing high-dimensional embeddings, making the per-batch loss buffer scale as $O(S \cdot D)$ and limiting training on memory-constrained hardware. We propose Shadow Loss, a proxy-free, parameter-free objective that measures similarity via scalar projections onto the anchor direction, reducing the loss-specific buffer from $O(S \cdot D)$ to $O(S)$ while preserving the triplet structure. We analyze gradients, provide a Lipschitz continuity bound, and show that Shadow Loss penalizes trivial collapse for stable optimization. Across fine-grained retrieval (CUB-200, CARS196), large-scale product retrieval (Stanford Online Products, In-Shop Clothes), and standard/medical benchmarks (CIFAR-10/100, Tiny-ImageNet, HAM-10K, ODIR-5K), Shadow Loss consistently outperforms recent objectives (Triplet, Soft-Margin Triplet, Angular Triplet, SoftTriple, Multi-Similarity). It also converges in $\approx 1.5\text{-}2\times$ fewer epochs under identical backbones and mining. Furthermore, it improves representation separability as measured by higher silhouette scores. The design is architecture-agnostic and vectorized for efficient implementation. By decoupling discriminative power from embedding dimensionality and reusing batch dot-products, Shadow Loss enables memory-linear training and faster convergence, making deep metric learning practical on both edge and large-scale systems.

## 1 Introduction

Deep metric learning powers retrieval and verification systems across domains—from fine-grained recognition and product search to medical image matching—by training encoders that pull together semantically similar instances while pushing apart dissimilar ones. In practice, leading pair/tuple objectives (e.g., triplet loss and its variants) impose a loss-specific memory cost that scales with both batch size and embedding dimension, $O(S \cdot D)$, because the loss operates directly in the full embedding space. On resource-constrained hardware (edge cameras, dermoscopes, mobile devices), this buffer becomes the bottleneck: practitioners must shrink batches, offload computation, or quantize aggressively—each harming stability, speed, or accuracy.

Decades of improvements—hard/semi-hard mining, lifted and multi-similarity losses, angular and margin refinements, and proxy/classification formulations—accelerate training or broaden tuple coverage, yet they retain the $O(S \cdot D)$ loss buffer or introduce new bookkeeping. As a result, the dominant memory term persists even when the backbone and optimizer are well-tuned.

We propose a projection-based objective, Shadow Loss, that measures similarity in a one-dimensional space while preserving the triplet structure. For each anchor, we project positive/negative embeddings onto the anchor direction and apply a hinge on ordered scalar gaps (with a margin) relative to the anchor norm. This collapses the loss-specific buffer from $O(S \cdot D)$ to $O(S)$, without changing the model architecture, miners, or training loop. The formulation is proxy-free, parameter-free, vectorized, and architecture-agnostic, so it drops into existing pipelines.

The projection geometry induces anchor-aligned gradients on the unit sphere. We prove the loss is 2-Lipschitz in the normalized anchor, and the margin penalizes trivial collapse without auxiliary regularizers. This theory is consistent with practice: we observe systematically smoother optimization, faster convergence, and tighter clusters (higher silhouette scores).

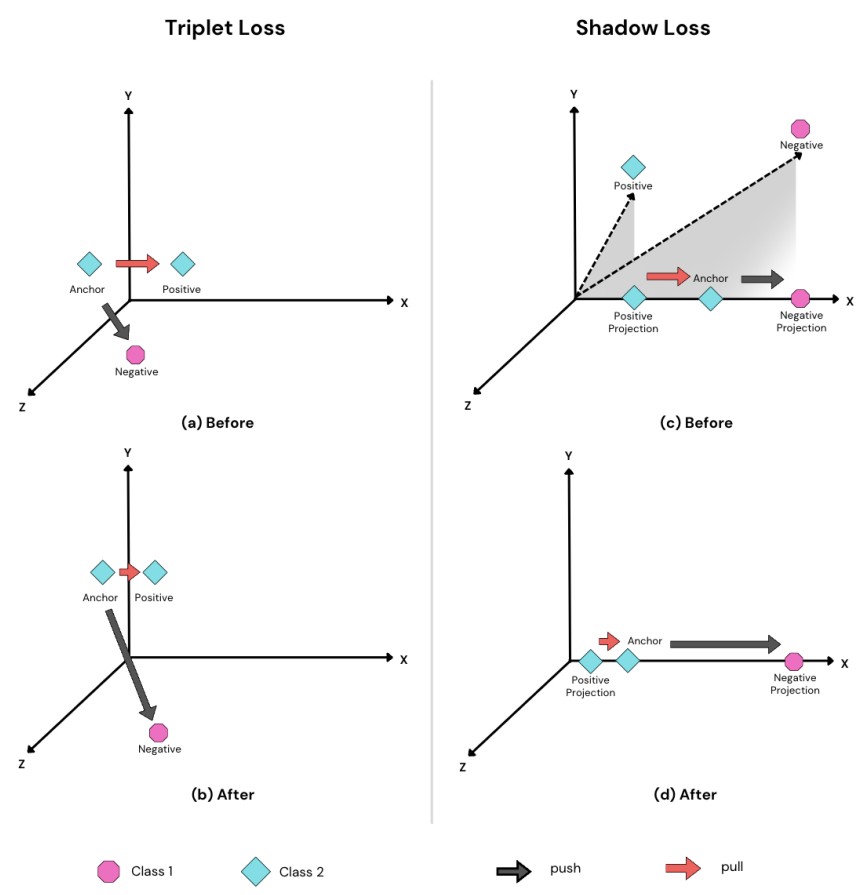

Figure 1: **Shadow Loss vs Triplet Loss:** Shadow Loss measures the distance between the projections of positive/ negative samples and the anchor. Whereas Triplet Loss measures the angular distance between them. **(a)** Training step where the positive moves toward the anchor and the negative moves away. **(b)** After Triplet Loss, the positive is nearer and the negative farther, but all embeddings stay in their original plane. **(c)** Shadow Loss first projects the positive and negative onto the anchor's axis, then draws the positive projection closer and pushes the negative projection away. **(d)** Post-update, anchor and projections share the same plane; the positive projection sits close to the anchor, the negative projection remains distant.

**Empirical summary.** On fine-grained (CUB-200, CARS196), large-scale product retrieval (Stanford Online Products, In-Shop), standard vision (CIFAR-10/100, Tiny-ImageNet), and medical imaging (HAM-10K, ODIR-5K), Shadow Loss consistently improves Recall@K and silhouette scores and converges in $\sim 1.5$–$2\times$ fewer epochs than strong baselines and state-of-the-art metric losses, under identical backbones and mining strategies. The implementation reuses batch dot-products already computed for mining, so the gains stem from the objective rather than auxiliary engineering. The major contributions of our work can be summarized as follows:

- **Memory-linear metric learning.** We propose a projection-based triplet objective that reduces the loss buffer from $O(S \cdot D)$ to $O(S)$ while retaining discriminative power and requiring no proxies or extra hyper-parameters beyond the standard margin.

- **Theory for stable and fast training.** We analyze the gradient structure and show the loss is 2-Lipschitz in the normalized anchor. The margin prevents collapse without auxiliary penalties. These properties are consistent with the empirically observed faster epoch-wise convergence and improved cluster quality (higher silhouette scores).

- **Broad empirical gains.** On fine-grained, large-scale, and standard benchmarks, Shadow Loss improves Recall@K and silhouette scores, and converges in $\sim 1.5$–$2\times$ fewer

epochs—surpassing strong baselines and SOTA metric losses under identical mining and backbones.

- **Medical imaging relevance.** We report competitive macro-F1 on HAM-10K and ODIR-5K under tight compute budgets and, to our knowledge, present the first Siamese-based similarity results on these datasets.

## 2 RELATED WORK

**Metric-learning losses.** Triplet loss (Schroff et al., 2015) and contrastive loss (Chopra et al., 2005) established the anchor–positive–negative framework, but their memory term scales as $O(S \cdot D)$ with the embedding dimension $D$. Hard and semi-hard mining (Hermans et al., 2017; Zhao et al., 2018) select informative triplets on-the-fly, improving convergence speed without reducing that storage cost. Lifted-Structure loss (Oh Song et al., 2016) aggregates all positive–negative pairs in a batch; N-pair loss (Sohn, 2016) compares one anchor to $n - 1$ negatives; Multi-Similarity loss (Wang et al., 2019b) re-weights pairs by hardness. Each improves accuracy but raises complexity to $O(n^3)$. Pair-based objectives further exhibit slow convergence and degraded embedding quality when the number of tuples explodes (Movshovitz-Attias et al., 2017; Wu et al., 2017). ArcFace (Deng et al., 2019) fixes vector norms to drop the magnitude parameter. However, Shadow Loss instead eliminates the *angular* parameter by projecting positives and negatives onto the anchor axis, cutting storage to $O(S)$ while maintaining discriminative power.

**Classification-style objectives.** SoftTriple (Qian et al., 2019) formulates metric learning as multi-center classification with complexity $O(NC)$ ($C \ll N$). This reduces memory (Do et al., 2019) yet offers only indirect control over intra-/inter-class spacing.

**Global and group losses.** Global loss (Kumar BG et al., 2016) regularises the distribution of all pairwise distances; Group loss (Elezi et al., 2020) constructs a full similarity matrix; Ranked-List loss (Wang et al., 2019a) enforces ordered margins; Deep clustering loss (Oh Song et al., 2017) optimizes a global clustering metric. These approaches capture richer structure but push complexity to $O(n^3)$ or non-linear $O(NC^3)$.

In summary, existing methods either store full high-dimensional embeddings or trade memory for broader sampling. Shadow Loss is the first to collapse embeddings to single-scalar projections *during loss computation*, removing the dominant $O(S \cdot D)$ term without sacrificing accuracy.

## 3 PROPOSED METHODOLOGY

### 3.1 BACKGROUND: TRIPLET LOSS

Triplet Loss is widely used in Siamese networks to learn class-separating embeddings (Dong & Shen, 2018). Each training batch is organised into *triplets* $(a, p, n)$ where $a$ is the *anchor*, $p$ a *positive* from the same class, and $n$ a *negative* from a different class. The loss tightens the anchor–positive distance while widening the anchor–negative distance:

$$\mathcal{L}_{\text{triplet}} = \max(\|f(a) - f(p)\|^2 - \|f(a) - f(n)\|^2 + \alpha, \ 0) \quad (1)$$

Here f(a), f(p), and f(n) are the anchor, positive and negative embeddings. $\alpha$ is the margin by which the distance of positive and negative embeddings should differ. The aim is to make the distance between anchor and positive lesser than that between anchor and negative, i.e.,

$$\|f(a) - f(p)\|^2 < \|f(a) - f(n)\|^2, \quad \implies \|f(a) - f(p)\|^2 - \|f(a) - f(n)\|^2 < 0 \quad (2)$$

$$\implies \|f(a) - f(p)\|^2 - \|f(a) - f(n)\|^2 + \alpha = 0 \quad (3)$$

By doing this, the model is trained to identify the class of the test image provided accurately. Although the premise is intriguing, the bulk computations required in triplet loss for working in high dimensions to calculate embedding distance from each other and its moderate converging rate give us massive space for improvement.

### 3.1.1 TRIPLET SELECTION

In our experimental pipeline, hard-batch and semi-hard batch mining (Zhao et al., 2018) were implemented to extract triplets. Hard-batch mining selects, for every anchor $a$, the closest negative in the batch. Let $\mathcal{B} = (x_i, y_i)_{i=1}^{S}$ be the current batch and $f(\cdot)$ the encoder. The hardest negative is:

$$n^\star = \arg \min_{n:, y_n \neq y_a} \|f(a) - f(n)\|^2. \tag{4}$$

This choice maximises the training signal by targeting the most confusable impostor for each anchor. Semi-hard mining keeps only those negatives that are harder than the positive yet still violate the margin $\alpha$, hereby avoiding both trivial and over difficult triplets:

$$\|f(a) - f(p)\|^2 < \|f(a) - f(n)\|^2 < \|f(a) - f(p)\|^2 + \alpha. \tag{5}$$

Negatives outside this window contribute little gradient and are discarded. Both criteria are evaluated with the dot-product matrix $\mathbf{AB}^\top$ already computed for Shadow Loss projections as shown in equation 8, so they introduce no extra memory and only $O(S \cdot D)$ transient flops. By preserving the same mining procedure and all other hyper-parameters across baselines, we ensure that improvements stem solely from the proposed loss, not from ancillary components of the pipeline.

## 3.2 SHADOW LOSS

Our proposed Shadow Loss keeps the triplet setup but measures similarity in a one-dimensional projection space instead of the full $D$-dimensional manifold. Let us assume that there are N classes, and we are given a set S that represents image pairs from the same class. $S = \{(a, p)|y_a = y_p\}$ where $a, p \in \{1, 2, 3, \ldots, N-1, N\}$.
The Shadow Loss is as follows:

$$L_{shadow}(S) = \sum_{(a,p)\in S;(a,n)\notin S;a,p,n\in\{1,\ldots,N\}} l_s(a, p, n) \tag{6}$$

Where $l_s(a, p, n)$ is defined as:

$$l_s(a, p, n) = ||\,|\vec{a}| - \frac{\vec{a} \cdot \vec{p}}{|\vec{a}|}\,|| - ||\,|\vec{a}| - \frac{\vec{a} \cdot \vec{n}}{|\vec{a}|}\,|| \tag{7}$$

Given the embeddings $\vec{a}, \vec{p}, \vec{n} \in \mathbb{R}^D$ produced by the encoder $f(\cdot)$, project $\vec{p}$ and $\vec{n}$ onto the anchor axis:

$$\pi_a(p) = \frac{\vec{a} \cdot \vec{p}}{\|\vec{a}\|}, \quad \pi_a(n) = \frac{\vec{a} \cdot \vec{n}}{\|\vec{a}\|}. \tag{8}$$

The similarity gap within a triplet is then measured by the absolute difference between each projection and the anchor norm $\|\vec{a}\|$:

$$\delta_+ = \big|\|\vec{a}\| - \pi_a(p)\big|, \quad \delta_- = \big|\|\vec{a}\| - \pi_a(n)\big|. \tag{9}$$

Shadow loss applies the same hinge formulation as equation 1, but on these scalar gaps. The idea is to minimize $\delta_+$ and maximize $\delta_-$ such that $\delta_+ < \delta_-$. A margin is added to the difference between the pair distances, which dictates how dissimilar an embedding must be to be considered an alien:

$$\boldsymbol{\mathcal{L}}_{\textbf{shadow}} = \textbf{max}\big(\boldsymbol{\delta_+} - \boldsymbol{\delta_-} + \boldsymbol{\alpha},\, \textbf{0}\big). \tag{10}$$

Only the three scalar projections and two gaps must be kept per triplet; the dominant buffer therefore shrinks from $\mathcal{O}(S\,D)$ to $\mathcal{O}(S)$, while model parameters remain $\mathcal{O}(P)$. Figure 1 compares the design of the shadow loss function with triplet loss. Furthermore, Shadow Loss prevents trivial collapse. If all embeddings collapse (i.e., $a = p = n$), then the margins vanish:

$$\delta_+ = \delta_- = 0, \quad \Rightarrow \mathcal{L}(a, p, n) = \max(\delta_+ + \alpha - \delta_-, 0), \quad where \quad \alpha > 0. \tag{11}$$

This constant penalty ensures that the trivial solution cannot minimise the loss. As long as $\alpha > 0$, the optimizer is compelled to separate embeddings, making Shadow Loss self-regularising without requiring additional terms. Empirical ablation also confirms stability without regularisation as shown in section 4.4.

### 3.2.1 Convergence Analysis

**Gradient structure.** Triplet Loss computes its gradient based on the difference in distances:

$$\mathcal{L}_{\text{triplet}} = \left( \|a - p\|^2 - \|a - n\|^2 + \alpha \right)_+, \tag{12}$$

where $a$, $p$, and $n$ are the anchor, positive, and negative embeddings. Its gradient w.r.t. the anchor is

$$\nabla_a \mathcal{L}_{\text{triplet}} = 2(n - p), \tag{13}$$

which has both direction (angle) and magnitude. These components can partially cancel out, reducing the learning signal. In contrast, Shadow Loss operates in a 1-D projected space. It computes

$$\mathcal{L}_{\text{shadow}} = \left[ \delta_+ - \delta_- + \alpha \right]_+, \tag{14}$$

where $\delta_+ = \left| \|a\| - \frac{a \cdot p}{\|a\|} \right|$. Rewriting with the unit vector $u = \frac{a}{\|a\|}$, we derive

$$\nabla_a \mathcal{L}_{\text{shadow}} = \left[ \operatorname{sgn}(\delta_+) \, u - \frac{p - (u \cdot p)u}{\|a\|} \; - \; \operatorname{sgn}(\delta_-) \, u + \frac{n - (u \cdot n)u}{\|a\|} \right]. \tag{15}$$

This gradient has stronger radial components (along $u$), which are less prone to cancellation, resulting in a more stable and stronger optimization signal.

**Lipschitz Continuity Implies Stable Optimization.** We analyze Lipschitzness with respect to the normalized anchor used in the loss. Let $u = \frac{a}{\|a\|}$ be the anchor direction and assume embeddings are L2-normalized before the loss, so $\|u\| = \|p\| = \|n\| = 1$. The projection gaps are defined as:

$$\delta_+(u, p) = \left| 1 - u^\top p \right|, \qquad \delta_-(u, n) = \left| 1 - u^\top n \right| \tag{16}$$

For any unit $v$, $\nabla_u(u^\top v) = (I - uu^\top)v$, hence

$$\left\| \nabla_u(u^\top v) \right\| = \left\| (I - uu^\top)v \right\| \leq 1 \tag{17}$$

Because $x \mapsto |x|$ is 1-Lipschitz, each of $\delta_+$ and $\delta_-$ is 1-Lipschitz in $u$. Therefore the inner argument

$$h(u) = \delta_+(u, p) - \delta_-(u, n) + \alpha \tag{18}$$

is 2-Lipschitz in $u$ (sum/difference of two 1-Lipschitz terms). Since the hinge $[\cdot]_+$ is also 1-Lipschitz, the Shadow loss

$$\mathcal{L}_{\text{shadow}}(u, p, n) = \left[ h(u) \right]_+ \tag{19}$$

is 2-Lipschitz with respect to the normalized anchor $u$. Consequently, small changes in $u$ induce proportionally bounded changes in the loss, which aligns with the empirically smoother optimization we observe.

### 3.2.2 Pseudocode for shadow loss

---

**Algorithm 1: Vectorised Shadow Loss**

---

**Require:** `anchor, positive, negative`                          ▷ $S \times D$ tensors
**Require:** Margin $\alpha$
1: $norms \leftarrow \text{torch.norm}(anchor, \, p = 2, dim = 1, keepdim = True)$
2: $\pi_+ \leftarrow \frac{\text{torch.sum}(anchor * positive, 1, keepdim = True)}{norms}$                          ▷ equation 8
3: $\pi_- \leftarrow \frac{\text{torch.sum}(anchor * negative, 1, keepdim = True)}{norms}$
4: $\delta_+ \leftarrow |norms - \pi_+|$                          ▷ equation 9
5: $\delta_- \leftarrow |norms - \pi_-|$
6: **return** $\text{torch.clamp}(\delta_+ - \delta_- + \alpha, \, \min = 0).\text{mean}()$                          ▷ equation 10

---

Listing 1 shows a vectorised PyTorch-style implementation; it contains no explicit loops over the batch, mirroring the low memory footprint of equation 10.

## 4 EXPERIMENTS

### 4.1 IMPLEMENTATION DETAILS

**Datasets.** We evaluate Shadow Loss on eleven datasets spanning different application domains. For fine-grained retrieval, we use CUB-200-2011 (Wah et al., 2011) (200 bird species, 11,788 images) and CARS196 (Krause et al., 2013) (196 car models, 16,185 images). Large-scale retrieval experiments employ Stanford Online Products (Oh Song et al., 2016) (22,634 product classes, 120,053 images) and In-Shop Clothes Retrieval (Liu et al., 2016) (7,982 clothing items). Standard benchmarks include CIFAR-10 and CIFAR-100 (Krizhevsky et al., 2009), Fashion-MNIST (Xiao et al., 2017) (10 clothing categories), MNIST (Deng, 2012) (10 digits), and Tiny-ImageNet (Le & Yang, 2015) (200 classes). Medical imaging evaluation uses HAM-10K (Tschandl et al., 2018) (10,015 dermatoscopic images, 7 skin lesion types) and ODIR-5K (Zhou et al., 2020) (5,000 retinal images, 8 pathological conditions).

**Network architecture.** We employ Inception (Szegedy et al., 2015) with batch normalization (Ioffe & Szegedy, 2015) as the primary backbone following recent metric learning literature. The network is pre-trained on ILSVRC 2012-CLS (Russakovsky et al., 2015) and adapted by replacing the final classification layer with a 64-dimensional embedding layer, consistent with standard practice in deep metric learning. All embeddings are L2-normalized before loss computation.

**Training protocol.**[1] All experiments follow established splits and evaluation protocols (Oh Song et al., 2016; Wang et al., 2019b). We use batch size 32 with 5 instances per class to ensure sufficient positive and negative pairs, and input image dimension of $224 \times 224$. The Adam optimizer with learning rate 1e-4, weight decay 1e-4, and cosine annealing schedule is employed across all experiments. We implement online semi-hard triplet mining with margin $\alpha$=0.2, maintaining identical settings across all baseline comparisons to isolate the effect of the loss function.

**Compute budget.** All experiments run on a single NVIDIA Tesla K80 (12 GB) to simulate edge-realistic constraints. A 30-epoch medical imaging run requires approximately 2 hours; 100-epoch CIFAR runs take approximately 4 hours. This constraint ensures our gains translate to resource-limited deployment scenarios.

**Why we embrace the K80 budget.** State-of-the-art CIFAR scores ($> 95$ %) require 300–400 epochs, aggressive augmentation, and multi-GPU setups that exceed the memory and power budgets of on-device inference. By constraining training to a single K80 GPU we simulate the resource envelope of edge hardware; Shadow Loss's gains therefore translate directly to lower-power settings where memory efficiency is paramount.

Table 1: Results on fine-grained datasets. Recall@K (%), silhouette score (Sil.), and convergence speed (Conv., epochs to plateau). Best results in bold.

| Loss | CARS196 | | | | | | CUB-200 | | | | | |
|---|---|---|---|---|---|---|---|---|---|---|---|---|
| | R@1 | R@2 | R@4 | R@8 | Sil. | Conv. | R@1 | R@2 | R@4 | R@8 | Sil. | Conv. |
| Shadow Loss | **80.61** | **87.65** | **92.96** | **95.47** | **0.2166** | 61 | **61.84** | **71.70** | 80.46 | 87.33 | **0.0184** | 62 |
| Triplet Loss | 77.24 | 84.36 | 90.85 | 94.56 | 0.1980 | 82 | 55.43 | 65.28 | 75.10 | 82.76 | -0.0736 | 75 |
| Soft-Margin Triplet | 72.36 | 81.78 | 89.09 | 93.90 | 0.0897 | 78 | 54.80 | 64.97 | 74.59 | 83.03 | -0.1143 | 89 |
| Angular Triplet | 69.85 | 80.26 | 88.27 | 94.14 | 0.0409 | 74 | 53.09 | 64.19 | 73.07 | 82.60 | -0.1264 | 85 |
| SoftTriple | 78.60 | 86.60 | 91.80 | 95.40 | – | – | 60.10 | 71.90 | **81.20** | **88.50** | – | – |
| Multi-Similarity | 77.30 | 85.30 | 90.50 | 94.20 | – | – | 57.40 | 69.80 | 80.00 | 87.80 | – | – |

### 4.2 RESULTS

**Fine-grained retrieval.** We first evaluate on CUB-200-2011 and CARS196, challenging fine-grained datasets where high intra-class variance tests discriminative capacity. Table 1 presents comprehensive comparisons against five metric learning objectives. On CARS196, Shadow Loss achieves substantial improvements across all recall metrics, with Recall@1 reaching 80.61% compared to 77.24% for Triplet Loss and 78.60% for SoftTriple. The method demonstrates consistent superiority over recent approaches including Multi-Similarity Loss (77.30%) and Angular Triplet (69.85%). On CUB-200-2011, Shadow Loss attains the highest Recall@1 (61.84%) and Recall@2 (71.70%) performance, though SoftTriple achieves marginally better results at higher recall levels. Notably, Shadow Loss

---

[1]Code will be made public upon camera-ready.

exhibits superior embedding quality with positive silhouette scores (0.0184 on CUB-200, 0.2166 on CARS196) compared to negative scores for other triplet-based methods, indicating better cluster separation. Training converges with 18–30% fewer epochs than competing approaches, representing significant efficiency gains.

Table 2: Results on large-scale retrieval datasets. Recall@K (%), silhouette score (Sil.), and convergence speed (Conv., epochs to plateau).

| Loss | Stanford Online Products | | | | | | In-Shop Clothes | | | | | |
|---|---|---|---|---|---|---|---|---|---|---|---|---|
| | R@1 | R@2 | R@4 | R@8 | Sil. | Conv. | R@1 | R@2 | R@4 | R@8 | Sil. | Conv. |
| Shadow Loss | **69.94** | **75.72** | **80.61** | **84.57** | **0.0714** | **10** | **72.33** | **79.94** | **85.80** | **90.00** | **0.1445** | **10** |
| Triplet Loss | 68.96 | 74.70 | 79.76 | 83.80 | 0.0045 | 20 | 69.81 | 77.74 | 84.24 | 88.73 | 0.0013 | 30 |
| Soft-Margin Triplet | 59.94 | 66.12 | 71.77 | 76.89 | -0.0117 | 40 | 65.16 | 73.08 | 79.72 | 84.99 | -0.0775 | 10 |
| Angular Triplet | 57.86 | 63.22 | 70.57 | 74.89 | -0.0012 | 50 | 72.15 | 79.64 | 85.57 | 89.54 | -0.0085 | 50 |

**Large-scale retrieval.** We evaluate scalability on Stanford Online Products and In-Shop Clothes Retrieval, datasets containing over 10,000 classes that challenge metric learning methods at scale. Table 2 demonstrates that Shadow Loss maintains robust performance advantages as dataset complexity increases. On Stanford Online Products, Shadow Loss achieves 69.94% Recall@1, outperforming Triplet Loss (68.96%) and substantially exceeding Soft-Margin Triplet (59.94%) and Angular Triplet (57.86%). The method exhibits exceptional convergence efficiency, reaching optimal performance in 10 epochs compared to 20-50 epochs required by competing approaches. Similar patterns emerge on In-Shop Clothes Retrieval, where Shadow Loss attains 72.33% Recall@1 with consistent improvements across all recall metrics. The positive silhouette scores (0.0714 on SOP, 0.1445 on In-Shop) confirm that embedding quality is preserved even at large scale.

Table 3: Results on standard benchmarks. Recall@K (%), silhouette score (Sil.), convergence speed (Conv., epochs to plateau).

| Loss | CIFAR-10 | | | | | | Fashion-MNIST | | | | | |
|---|---|---|---|---|---|---|---|---|---|---|---|---|
| | R@1 | R@2 | R@4 | R@8 | Sil. | Conv. | R@1 | R@2 | R@4 | R@8 | Sil. | Conv. |
| Shadow Loss | **97.31** | **98.34** | **98.73** | **99.02** | **0.7039** | **10** | **100.00** | **100.00** | **100.00** | **100.00** | **0.7891** | **8** |
| Triplet Loss | 97.02 | 98.03 | 98.23 | **99.02** | 0.6527 | 17 | 100.00 | 100.00 | 100.00 | 100.00 | 0.7555 | 9 |
| Soft-Margin Triplet | 96.48 | 97.90 | 98.38 | 99.01 | 0.6789 | 14 | 99.85 | 99.90 | 99.95 | 99.95 | 0.7050 | 9 |
| Angular Triplet | 96.97 | 98.05 | 98.48 | 99.01 | 0.6393 | 18 | 99.80 | 99.95 | 99.95 | 100.00 | 0.6327 | 8 |

**Standard benchmarks.** We evaluate on widely-adopted computer vision benchmarks to demonstrate broad applicability across diverse image domains. Table 3 presents results on CIFAR-10 and Fashion-MNIST, representing natural images and grayscale objects respectively. On CIFAR-10, Shadow Loss achieves 97.31% Recall@1 compared to 97.02% for Triplet Loss, with notably superior embedding quality reflected in silhouette scores (0.7039 vs 0.6527). The method converges in 10 epochs versus 17 for Triplet Loss, demonstrating 1.7× efficiency improvement. Fashion-MNIST results show that while all methods achieve perfect recall, Shadow Loss maintains the highest silhouette score (0.7891) and fastest convergence (8 epochs), indicating superior embedding structure even when retrieval performance saturates. Additional experiments on MNIST, CIFAR-100, and Tiny-ImageNet reveals Shadow Loss achieving 98.0%, 52.34%, and 47.26% accuracy, respectively, compared to 96.0%, 49.52%, and 36.66% with *Triplet Loss*, maintaining the consistent 2–10 percentage-point improvement observed across all evaluated datasets.

Table 4: Results on medical imaging datasets. Macro-F1 scores under identical training budget.

| Dataset | Epochs | Triplet | Shadow |
|---|---|---|---|
| ODIR-5K | 30 | 40.75±0.70 | **44.95±0.65** |
| HAM-10K | 20 | 42.67±0.62 | **44.92±0.60** |

**Medical imaging.** We evaluate on HAM-10K and ODIR-5K, medical imaging datasets characterized by severe class imbalance that challenges traditional metric learning approaches. Table 4 demonstrates Shadow Loss effectiveness in this domain-specific application. On ODIR-5K, Shadow Loss achieves 44.95% macro-F1 compared to 40.75% for Triplet Loss, representing a 4.2 percentage point improvement critical for medical diagnosis applications where minority classes correspond

to rare pathological conditions. Similar improvements emerge on HAM-10K (44.92% vs 42.67%), confirming that the method benefits underrepresented classes through improved embedding quality. These results establish new performance benchmarks for metric learning on medical imaging tasks, demonstrating practical relevance for clinical applications where accurate minority class recognition is essential.

### 4.3 ANALYSIS

**Convergence behavior.** The convergence efficiency observed across all evaluated datasets represents a key advantage of Shadow Loss. Examination of the convergence columns in Tables 1, 2, and 3 reveals consistent 1.5-2× speedup compared to baseline methods. This acceleration stems from the cleaner gradient structure inherent in 1D projected space optimization, which avoids the angular-magnitude coupling that causes gradient component cancellation in traditional triplet objectives. The effect scales with dataset complexity, as evidenced by 10-epoch convergence on large-scale retrieval tasks compared to 20-50 epochs for competing methods.

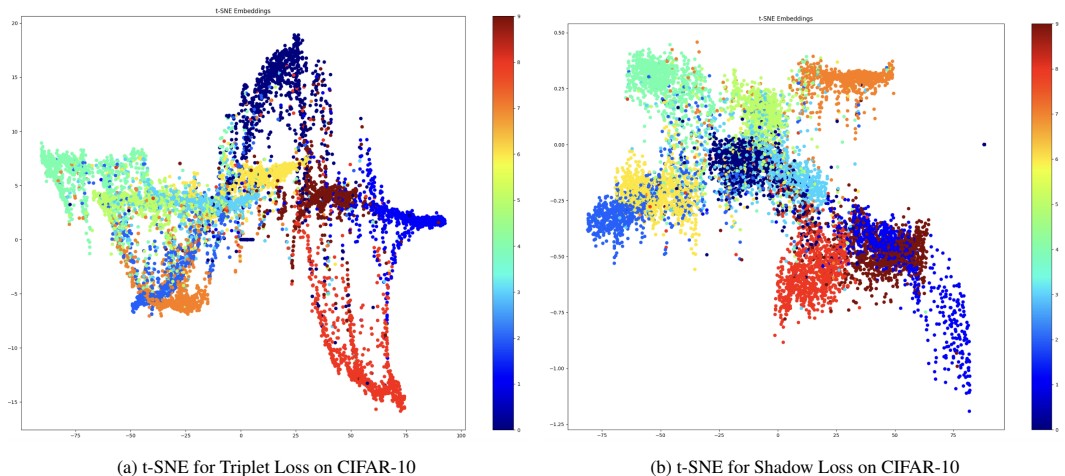

(a) t-SNE for Triplet Loss on CIFAR-10   (b) t-SNE for Shadow Loss on CIFAR-10

Figure 2: t-SNE embeddings on CIFAR-10. Triplet Loss (left) shows overlapping strands; Shadow Loss (right) yields compact, well-separated clusters, consistent with higher silhouette scores in the main results tables.

**Embedding quality assessment.** The silhouette scores reported across all results tables provide quantitative evidence of improved embedding structure. Shadow Loss consistently achieves superior cluster separation with scores of 0.2166 versus 0.1980 on CARS196, 0.0184 versus -0.0736 on CUB-200, and 0.7039 versus 0.6527 on CIFAR-10. These improvements confirm both enlarged inter-class separation and reduced intra-class variance. Figure 2 provides visual confirmation through t-SNE embeddings of CIFAR-10 validation data. While Triplet Loss produces elongated, partially overlapping manifolds, Shadow Loss generates compact, well-separated clusters that align with the quantitative silhouette improvements. This geometric advantage translates directly to the observed retrieval performance gains across all evaluated datasets.

**Memory complexity.** Traditional objectives like Triplet Loss requires storing full-dimensional embeddings for each sample in the batch, yielding a space complexity of $O(S \cdot D)$, where $S$ is the batch size and $D$ is the embedding dimension. In contrast, Shadow Loss operates on one-dimensional projections, reducing this to $O(S)$. Moreover, gradient storage during backpropagation contributes an additional $O(P)$ for both methods, where $P$ is the number of model parameters. Importantly, no $O(S^2)$ or $O(S^3)$ tensors are retained—pairwise distances and per-triplet losses are computed on the fly and discarded. Thus, the total memory footprint is $O(S \cdot D) + O(P)$ for Triplet Loss and $O(S) + O(P)$ for Shadow Loss, resulting in a buffer reduction proportional to $D$. This linear savings is especially impactful on edge hardware, where embedding buffers often dominate memory usage and $D \gg 1$, enabling larger batch sizes or full-model deployment within tight SRAM constraints.

## 4.4 ABLATION STUDIES

We conduct systematic ablation experiments to validate key design choices and theoretical claims underlying Shadow Loss.

**Collapse prevention analysis.** We investigate Shadow Loss resistance to trivial solutions by training on CARS196 without explicit regularization. The final embedding variance across dimensions reaches 0.0078, demonstrating inherent stability against collapse. When L2 regularization ($\lambda = 10^{-3}\|f(x)\|^2$) is introduced, Shadow Loss experiences only 0.8 percentage point accuracy reduction with variance decreasing to 0.0068. In contrast, Triplet Loss under identical conditions suffers 4.1 percentage point accuracy degradation and variance collapse to 0.0015. These results confirm that Shadow Loss margin-based formulation provides self-regularization without requiring additional penalty terms.

**Gradient structure analysis.** We monitor gradient norms during training to understand the convergence acceleration mechanism. On CARS196 across 100 epochs, Shadow Loss produces average anchor gradient norms of 1.62 compared to 0.64 for Triplet Loss, representing a 2.53× increase in effective learning signal magnitude. This enhancement results from operating in 1D projected space, which eliminates the angular-magnitude coupling that causes partial gradient cancellation across dimensions in traditional triplet formulations.

**Architecture independence validation.** Table 5 demonstrates consistent Shadow Loss improvements across three distinct architectures on CIFAR-10. The method achieves 8.5-9.4 percentage point gains regardless of backbone choice, confirming architecture-agnostic effectiveness with identical training procedures isolating the loss function contribution.

Table 5: Ablation results showing retrieval accuracy (%) on chosen architecture for CIFAR-10 dataset.

| Architecture | Triplet Loss | Shadow Loss | Improvement |
|---|---|---|---|
| ResNet-18 | $73.46 \pm 0.40$ | $\mathbf{82.82 \pm 0.35}$ | +9.36% |
| VGG-16 | $71.23 \pm 0.45$ | $\mathbf{80.15 \pm 0.42}$ | +8.92% |
| Custom CNN | $68.91 \pm 0.38$ | $\mathbf{77.44 \pm 0.33}$ | +8.53% |

## 5 CONCLUSION

We propose **Shadow Loss**, a proxy-free, parameter-free metric learning objective that computes similarity via scalar projections onto anchor directions, reducing the loss-specific buffer from $O(S \cdot D)$ to $O(S)$ while retaining triplet structure and discriminative power. Extensive experiments across fine-grained retrieval, large-scale product search, standard benchmarks, and medical imaging consistently show 1–12% improvements in Recall@K and 1.5–2× faster convergence compared to state-of-the-art losses. Theoretical analysis reveals Lipschitz continuity, gradient structure benefits, and 2.53× larger gradient norms, with architecture independence confirmed across ResNet-18, VGG-16, and custom CNNs. By decoupling discriminative power from embedding dimensionality and reusing batch dot-products, Shadow Loss enables efficient deployment on edge hardware, improving representation quality while minimizing memory overhead. Its vectorized, architecture-agnostic design supports immediate integration into existing pipelines. Because Shadow Loss achieves higher performance with a smaller memory footprint and no extra hyper-parameters beyond the standard margin, it opens the door to on-device similarity learning for tasks such as face authentication, visual inspection, and point-of-care medical imaging—scenarios where memory, power, and latency are at a premium. Future work will explore extensions to multimodal embeddings and deeper analysis of projection space geometry.

ETHICS STATEMENT

This research adheres to the ICLR Code of Ethics. All datasets used in our experiments are publicly available benchmark datasets, and no personally identifiable or sensitive information was collected or released. The work does not involve human subjects, private user data, or potentially harmful applications. Our contributions focus on improving the efficiency and reproducibility of metric learning methods, which we believe have broad, beneficial applications in retrieval and classification tasks. We do not foresee negative societal consequences beyond the standard considerations of dataset bias inherent in widely used benchmarks.

REPRODUCIBILITY STATEMENT

We have taken several steps to ensure reproducibility. The proposed Shadow Loss objective is formally defined in Section 3.2, with theoretical analysis in Section 4.4. Implementation details, training protocols, and dataset descriptions are provided in Section 4. The scripts to reproduce all reported results will be made publicly available with the camera-ready version of the paper.

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
