# OpenReview forum: "Shadow loss: Memory-linear deep metric learning with anchor projection"
_ICLR.cc/2026/Conference — ICLR 2026 Conference Withdrawn Submission_

### Official Review · Reviewer_s8Ub · 2025-10-30

**Soundness:** 2
**Presentation:** 3
**Contribution:** 2
**Rating:** 2
**Confidence:** 4

**Summary:**

The paper proposes novel deep metric learning loss that focuses on memory efficiency to facilitate training on edge devices. The method extends the established triplet loss by projecting the positive and negative sample onto the anchor direction. This allows to reduce the loss memory buffer to scalar values instead of storing full embeddings for the triplet. The authors present several other desirable properties of the loss, such as its Lipschitz continuity, representation separability, and faster convergence. The efficacy of the loss is experimentally verified in a constrained setting with limited compute budget to simulate capabilities of edge devices. Standard fine-grained retrieval benchmarks are used to evaluate and compare the trained model against several well known losses. Additionally, the method is evaluated on two medical imaging datasets that pose a realistic use case.

**Strengths:**

**S1)** The paper is well structured, easy to read and understand. The proposed loss is described concisely and clearly using correct mathematical notation.

**Weaknesses:**

**W1)** The loss follows a standard practice and relies on triplet mining step. The memory bottleneck that was eliminated in the loss is still present in this step since all the embeddings of the batch have to be stored at once. This contradicts the whole motivation of the proposed method.

**W2**) The authors mention on line 286 that "*All embeddings are L2-normalized before loss computation.*" This is a crucial detail which appears to be inconsistent with several parts of the paper. If the embeddings are truly L2-normalized, as is the standard practice in deep metric learning, then the loss would be drastically simplified. The Convergence analysis section (3.2.1) would not hold since the gradient would be proportional to the one of a triplet loss, invalidating the argument. Even if the embeddings are not L2-normalized, the discussion in 3.2.1 is still misleading as the gradient depends on the unnormalized embeddings and is influenced by their magnitudes.

**W3)** It is not mentioned how the hyperparameters of the losses are selected. The margin used in triplet loss is a key component and its optimal setting typically varies significantly depending on the training set domain. For a fair comparison between the compared methods, it is necessary to discuss the selection of these parameters.

**W4**) Since the main contribution of the paper is introducing a small tweaks to the established triplet loss, the experiments should cover more comparisons. Even though Table 5 includes other backbones than just the default Inception, it is missing other loss functions. A stronger state-of-the-art networks, such as ViT, should be included to confirm gains of the loss. I would also suggest using other datasets than just CIFAR-10, which is too easy. All the compared losses are more than 5 years old. Comparing to more recent losses such as \[1\] would be beneficial.

[1] Recall@k Surrogate Loss with Large Batches and Similarity Mixup, Patel et al., CVPR 2022


Minor weaknesses:

MW1) The motivation of the loss could be discussed further and extended with examples of some practical use cases. I find it difficult to imagine a setup where supervised training is conducted on edge device and the model is then subsequently used to perform a ranking of a pre-existing database.

MW2) Lipschitz continuity (3.2.1) discussion is unnecessary for the main paper and can be moved to the supplementary material to improve the flow of the method presentation. The shadow loss is a simple extension of a triplet loss with the projection gaps, and as such it is 2-Lipschitz in the same manner. Furthermore, the implications of Lipschitz continuity of the loss on the final performance are not generally clear.

MW3) Collapse prevention analysis in 4.4 is unclear. Please clarify the difference between the effect of the margin in triplet loss compared to the margin used in shadow loss. The measured metrics like "point accuracy" need more explanation.

MW4) **A** and **B** are not defined prior to their use on line 175.

MW5) Fashion-MNIST results are useless as all the method reach (almost) perfect retrieval.

**Questions:**

Q1) How are the hyperparameters for all the methods selected?

Q2) Why are the SoftTriple and Multi-Similarity loss appearing in Table 1 but are missing in Table 2 and Table 3?

---

### Official Review · Reviewer_oUht · 2025-10-31

**Soundness:** 2
**Presentation:** 2
**Contribution:** 2
**Rating:** 4
**Confidence:** 3

**Summary:**

The paper proposes a proxy-free, parameter-free triplet-style Shadow Loss  that computes the loss in 1-D by projecting positive/negative embeddings onto the anchor direction. The promise is: (i) loss-side memory shrinks from O(S·D) → O(S), (ii) cleaner gradients  and (iii) faster convergence with better Recall@K and higher silhouette scores across fine-grained (CUB, CARS), “standard” computer vision datasets (CIFARs, Tiny-ImageNet), and two medical datasets (HAM-10K, ODIR-5K).

**Strengths:**

1.Shadow Loss preserves triplet semantics but reduces the loss-specific buffer to O(S); the formulation is proxy-free and parameter-free (beyond a margin). That could be attractive for edge-based deployments and other resource constraint scenarios.

2. Covering traditional Computer Vision datasets and two medical datasets under constrained hardware advocates its generalization capability.

**Weaknesses:**

1. On projecting positives/negatives onto Anchor(a)  may lead to situation which ignores components orthogonal to ANchor.  A negative that’s far from (a) in angle but shares a similar projection can slip through the margin? May be this could be the case when intra-class manifolds are curved.

2. The proposed method has been compared against commonly used losses like Triplet, Soft-Margin Triplet, Angular Triplet, SoftTriple, Multi-Similarity —  but Proxy-Anchor, Circle Loss, and Supervised Contrastive (SupCon) are notably absent. These are strong, widely-used losses and would sharpen the claim of the paper if those are included in the comparison.

3. Seems that all experiments were conducted with fixed D=64 with L2 normalization. It would strengthen the experimental framework and also the claim of the method - if all experiments could be conducted  with  multiple embedding sizes (e.g., D = 64, 128, 256, 512), to confirm that the performance improvements hold consistently beyond this single setup.

**Questions:**

Could you show actual VRAM usage and achievable batch-size increases?

---

### Official Review · Reviewer_xVi6 · 2025-11-01

**Soundness:** 2
**Presentation:** 2
**Contribution:** 2
**Rating:** 4
**Confidence:** 3

**Summary:**

The concept of "Shadow Loss" involves projecting high-dimensional embeddings into a lower-dimensional projection space, where distances reflect class similarity. This approach facilitates faster convergence and reduces memory usage. The primary goal is to decrease both memory and computational costs while maintaining or improving the quality of retrieval and embedding. Shadow Loss employs anchor projection, which involves projecting embeddings onto a smaller-dimensional subspace through linear projection for distance computations. Additionally, it utilizes a memory-linear approach, meaning that a memory bank is created to store previous embeddings or anchors. Linear combinations are then used to define the anchors or the projection space.

**Strengths:**

1. Due to the memory-efficient projection combined with anchor and linear mechanisms, this method claims to converge faster than traditional metric losses like Triplet. This results in reduced training time and resource usage.

2. The “memory-linear” aspect indicates that the size of the memory bank, the anchor set, or the projection mechanism increases linearly rather than quadratically or cubically with the size of the dataset. This is an advantage for large-scale tasks.

**Weaknesses:**

1. The combination of anchor projection, a memory bank, and linear projection can introduce complexity. Implementations may require careful tuning of hyperparameters (e.g., projection dimension, anchor update rule, and memory bank size), which the paper does not detail.
2. If the underlying backbone embeddings are weak or poorly pretrained, projecting them into a reduced-dimensional space may amplify errors or lead to the collapse of the embedding.
3. Reducing the embedding dimension through linear projection carries the risk of losing representational power. A smaller embedding space may lead to greater overlap between different classes. Therefore, the assertion that it is possible to achieve “fewer dimensions with the same power” needs robust empirical evidence, and it may not be universally applicable.
4. Choosing a task with extremely fine-grained classes (such as face recognition involving many classes) might negate the benefits of memory savings when reducing dimensionality.
5. The number of datasets and their scale may be modest, meaning that a "5–10% accuracy improvement" could apply only to specific tasks or dataset categories. Generalization to larger benchmarks, such as extensive retrieval tasks, has yet to be demonstrated.

**Questions:**

1. How sensitive is Shadow Loss to hyperparameters such as projection dimension, memory bank size, and anchor update frequency?
2. Please elaborate on how the anchor projection is designed? Is it fixed, learned, or adaptively updated during training?
3. What happens when the backbone embeddings are weak or untrained? Does the projection amplify noise in the early training stages?
4. What’s the theoretical or empirical justification for claiming equivalent discriminative power in reduced-dimensional space?
5. Is it able to include a thorough ablation study to dissect the impact of: Projection dimension, Memory bank size, and Update frequency or strategy?

---

### Official Review · Reviewer_axFD · 2025-11-03

**Soundness:** 1
**Presentation:** 1
**Contribution:** 1
**Rating:** 0
**Confidence:** 5

**Summary:**

The authors propose a shadow loss objective for deep metric learning, which they claim to outperform recent objectives.

They claim that this loss converges 1.5-2x fewer epochs than baseline loss functions.

The new loss function additionally is proxy-free (does not rely on a classification objective).

The authors give some mathematical explanation for why the loss function behaves nicely in terms of gradient-based optimization.

**Strengths:**

- There is some mathematical justification

**Weaknesses:**

There are a few reasons I do not think this paper is publication-ready.

**Experimental**:
 - There are many missing baselines. The baselines the authors use are all quite dated.
 - Here is a library that implements some baselines: https://github.com/KevinMusgrave/pytorch-metric-learning
 - The authors do not compare against proxy-based losses, this is unacceptable.
 - There should be more benchmark datasets, especially more modern ones.

**Method**:
 - Section 3.2.1 is too hand-wavy. These should be stated as Theorems or propositions.

**Presentation**:
- What does O(SD) mean in the abstract? It is not advisable to use notation that has not yet been introduced.
- Section 3: There are many type setting errors for the math:
  - arg min is one word
  - Eq 7: What is the absolute value of a vector?
  - Vectors should be boldfaced to save space.
  - Many more ...

**Questions:**

None

---

### Note · Authors · 2025-11-14

I have read and agree with the venue's withdrawal policy on behalf of myself and my co-authors.